# Genetic and Pathogenic Characterization of a New *Iridovirus* Isolated from Cage-Cultured Large Yellow Croaker (*Larimichthys crocea*) in China

**DOI:** 10.3390/v14020208

**Published:** 2022-01-21

**Authors:** Gengshen Wang, Yingjia Luan, Jinping Wei, Yunfeng Li, Hui Shi, Haoxue Cheng, Aixu Bai, Jianjun Xie, Wenjun Xu, Pan Qin

**Affiliations:** 1Key Laboratory of Mariculture and Enhancement of Zhejiang Province, Marine Fisheries Research Institute of Zhejiang, Zhoushan 316100, China; wgs-1988@163.com (G.W.); huishi2002@126.com (H.S.); chx1504098365@163.com (H.C.); xiejianjun611@163.com (J.X.); xwenjun@sina.com (W.X.); 2Marine and Fisheries Research Institute, Zhejiang Ocean University, Zhoushan 316100, China; 3Key Laboratory of Marine Biotechnology of Fujian Province, College of Marine Sciences, Fujian Agriculture and Forestry University, Fuzhou 350002, China; 1301250069@163.com (Y.L.); wei197509@163.com (J.W.); mortimer2022@163.com (Y.L.); 4Huaian Customs District, Huaian 223001, China; bax871017@163.com

**Keywords:** large yellow croaker, iridovirus, genomic, pathogenicity

## Abstract

*Iridoviruses* are an important pathogen of ectothermic vertebrates and are considered a significant threat to aquacultural fish production. Recently, one of the most economically important marine species in China, the large yellow croaker *(Larimichthys crocea*), has been increasingly reported to be the victim of iridovirus disease. In this study, we isolated and identified a novel iridovirus, LYCIV-ZS-2020, from cage-cultured large yellow croaker farms in Zhoushan island, China. Genome sequencing and subsequent phylogenetic analyses showed that LYCIV-ZS-2020 belongs to the genus *Megalocytivirus* and is closely related to the Pompano iridoviruses isolated in the Dominican Republic. LYCIV-ZS-2020 enriched from selected tissues of naturally infected large yellow croaker was used in an artificial infection trial and the results proved its pathogenicity in large yellow croaker. This is the first systematic research on the genetic and pathogenic characterization of iridovirus in large yellow croakers, which expanded our knowledge of the iridovirus.

## 1. Introduction

The *Iridoviridae* family comprises large DNA viruses that either possess enveloped or nonenveloped virions measuring 120–350 nm in diameter, with linear and double-stranded DNA genomes ranging from 105 to 212 kbp [1]. Currently, members of the *Iridoviridae* family are classified into two subfamilies, which are *Alphairidovirinae* and *Betairidovirinae*, and six genera, namely *Iridovirus*, *Chloriridovirus*, *Decapodiridovirus*, *Lymphocystivirus*, *Megalocytivirus*, and *Ranavirus* (https://talk.ictvonline.org/taxonomy/, accessed on 8 December 2021). The *iridoviridae* family possesses broad species tropism and infects invertebrates [2] and poikilothermic vertebrates, such as fish [3,4], amphibians [5,6], and reptiles [5,6,7,8]. Among the six genera of *Iridoviridae*, *Ranavirus*, *Lymphocystivirus*, and *Megalocytivirus* were shown to cause infections in freshwater and marine fish worldwide and are associated with high morbidity and mortality [3,9,10]. Despite the economic and ecological significance of iridoviruses, very little is known regarding their molecular biology, origin, and evolutionary dynamics since there are limited data presenting the complete length genome of these viruses. 

Large yellow croaker (*Larimichthys crocea*) is distributed in Southeastern China on the coastal area and is one of the most economically important marine species artificially bred over the past three decades. Although artificial breeding greatly improved productivity in croaker rearing, aquacultured croakers have reduced genetic diversity and disease resistance, resulting in increased vulnerability to various marine pathogens [11], such as parasites [12], bacteria [13] and viruses [14]. 

During the summers of 1999–2001, a novel infectious disease with clinical signs including anorexia, abnormal swimming, darkened color, and gulping occurred in cultured large yellow croakers in the Fujian Province of China. Large numbers of virus particles with a diameter of 140–160 nm were present in the spleen and kidney of diseased fish. Further molecular analysis confirmed the pathogen was a novel iridovirus, which was later named large yellow croaker iridovirus (LYCIV) [15]. Unfortunately, serial passage of LYCIV in BF-2 cells resulted in a gradual decrease in infectivity and loss of infectious virus, which hinders the research work from going deeper.

In recent years, outbreaks of LYCIV in large yellow croakers have been increasingly reported in the field. As the virus failed to grow in commonly used cell lines, the pathogenicity and biological characteristics of LYCIV remain to be elucidated. Some researchers argue that LYCIV is the causative agent of “white gill disease”, which caused mass mortality of the cage-reared large yellow croakers. However, others consider that this virus is only a co-infectious agent which causes disease in large yellow croakers with other pathogens. 

In 2020, there was an observation of high mortality of fish larvae with signs such as black body coloration, white gills, and spleen and kidney enlargement in three cage-cultured large yellow croaker farms in Zhoushan island, China. Initial screening of common pathogens such as *Pseudomonas plecoglossicida*, *Aeromonas hydrophila*, *Vibrio alginolyticus*, *Vibrio harveyi*, and parasite all showed negative results. In this study, we identified a new iridovirus in these fish through transmission electron microscopy (TEM) and next-generation sequencing. Subsequent phylogenetic analysis and pathogenesis studies were also conducted. To our knowledge, this is the first systematic research on the genetic and pathogenic characterization of this novel iridovirus, which provided valuable experimental data for the study of its pathogenicity.

## 2. Materials and Methods

### 2.1. Sample Collection

Sampling was performed at the above-mentioned cage-reared large yellow croaker farm in Zhoushan island, China. Tissue samples of the gills, spleen, and kidney were collected from fish exhibiting obvious symptoms and subsequently used for electron microscope, DNA extraction and pathogenicity study. 

### 2.2. Electron Microscope

The samples of tissues were first fixed with 2.5% glutaraldehyde in phosphate buffer (0.1 M, pH 7.0) (Sinopharm, Beijing, China) for more than 4 h followed by a second post-fixation for 1 h in 1% osmium tetroxide in phosphate buffer for 2 h and washed three times in the phosphate buffer (0.1 M, pH 7.0) for 15 min each time. Subsequently, fixed tissues were dehydrated by a graded series of ethanol (30%, 50%, 70%, 80%) and acetone (90% and 95%) (Sinopharm, Beijing, China) for about 15 min each. The samples were then dehydrated twice by absolute acetone for 20 min each. After that, the specimens were placed in a 1:1 mixture of absolute acetone and final Spurr resin mixture (SPI Supplies, West Chester, PA, USA) for 1 h at room temperature, then transferred to a 1:3 mixture of absolute acetone and final resin mixture for 3 h, then kept overnight in final Spurr resin mixture. Finally, the specimens were sectioned in LEICA EM UC7 ultratome, stained by uranyl acetate and alkaline lead citrate (SPI Supplies, West Chester, PA, USA) for 10 min each, and examined with Hitachi Model H-7650 TEM.

### 2.3. Extraction of Viral Genome DNA

With some modifications, the Chen et al. method [15] was used to extract DNA in the spleen tissue samples from the naturally infected or healthy large yellow croakers. Briefly, spleen samples were homogenized in 1 mL of sterile redistilled water on ice. After centrifugation at 1500× *g* for 10 min at 4 °C, the supernatants were mixed with an equal volume of lysis buffer (10 mM Tris–HCl, pH 8.0, 100 mM EDTA, 20 g/mL RNase, 0.5% (*w*/*v*) SDS). Proteinase K was then added to reach 100 g/mL final concentration. The mixtures were incubated for 3 h at 55 °C and then extracted twice with phenol-chloroform. DNA was precipitated with ethanol for 30 min at −20 °C, then redissolved in 100 μL TE buffer (10 mM Tris–HCl, 1 mM EDTA, pH 8.0) and stored at −80 °C for further use. Same method was also used to extract viral genome DNA from virus inoculum used in animal experiments described in later section.

### 2.4. Virus Detection

The DNA samples were utilized for virus detection using qPCR method with specific primers and probe targeting the LYCIV ORF 21 gene (the forward primer 5′-CTGAGGGTGGTCGTCTGGTT-3′, the reverse primer 5′-ATGGCGACCCTGCTACTTCT-3′ and probe 5′-FARM-AAGGTGGTGGCGTGAGTACACGCCA-BHQ1-3′). To quantitate viral DNA, a standard curve was obtained using dilutions of a known quantity of the pEasy blunt-ORF21 plasmid DNA (determined spectrophotometrically). Seven consecutive dilutions (dilution factor 1:10) were prepared containing from 10^9^ to 10^3^ copies/reaction. The amounts of LYCIV DNA in tissue samples were obtained by plotting Ct values onto the standard curve. Cycle threshold value of 35 was considered as the limit of detection based upon validation data using the DNA standards.

### 2.5. Genome Sequencing, Assembly, and Annotation

Genomic DNA was quantified by using a TBS-380 fluorometer (Turner BioSystems Inc., Sunnyvale, CA, USA). DNA samples with high quality (i.e., OD260/280 = 1.8~2.0, >6 μg) are utilized to construct a fragment library. Sequencing was performed by Shanghai Biozeron Biotechnology Co., Ltd. (Shanghai, China). For Illumina pair-end sequencing of each strain, at least 1 μg genomic DNA was used for sequencing library construction. Paired-end libraries with insert sizes of ~400 bp were prepared following Illumina’s standard genomic DNA library preparation procedure. Purified genomic DNA was sheared into smaller fragments with the desired size by covaris, and blunt ends were generated using T4 DNA polymerase. After adding an “A” base to the 3′ end of the sharp phosphorylated DNA fragments, adapters were ligated to the ends of the DNA fragments. The desired fragments were purified through gel-electrophoresis, then selectively enriched and amplified by PCR. The index tag was introduced into the adapter at the PCR stage followed by a library-quality test. The qualified Illumina pair-end library was then used for sequencing with the Illumina NovaSeq 6000 system (https://www.illumina.com/systems/sequencing-platforms/novaseq.html, accessed on 15 November 2020).

The raw paired-end reads were trimmed and quality controlled by Trimmomatic (version 0.36) with sliding window set at 4:15 and minimum length at 75 bp (http://www.usadellab.org/cms/?page=trimmomatic, accessed on 15 November 2020). Clean data obtained by the above quality control processes were analyzed further. We used ABySS (http://www.bcgsc.ca/platform/bioinfo/software/abyss, accessed on 15 November 2020) to do genome assembly with multiple-Kmer parameters and the optimal results of the assembly were then inputted into the GapCloser software (https://sourceforge.net/projects/soapdenovo2/files/GapCloser/, accessed on 15 November 2020) which filled up the remaining local inner gaps and corrected the single base polymorphism for the final assembly results.

Annotation of the predicted ORFs was performed using Genome Annotation Transfer Utility (http://athena.bioc.uvic.ca/help/tool-help/help-books/genome-annotation-transfer-utility-gatu-documentation/, accessed on 20 November 2020) with the Pompano iridovirus genome isolate PIV2010 (GenBank accession no. MK098185) as a reference.

### 2.6. Genome Analysis

The full-length genome of the virus being studied and other representative strains of the *Iridoviridae* were aligned using MAFFT online version (https://mafft.cbrc.jp/alignment/software/, accessed on 20 April 2021). The phylogenetic analyses were performed using the neighbor-joining method in MEGA6.

### 2.7. Expression of Recombinant Major Capsid Protein (MCP) and Generation of Anti-MCP Polyclonal Antibodies (pAb) 

The major capsid protein (MCP) gene was cloned into the pET-28a (+) vector and pcDNA3.1, respectively. The recombinant plasmids were verified by DNA sequencing. The pET-28a-MCP recombinant plasmid was transformed into *Rosetta (DE3)* competent cells and induced by one mM IPTG for 5 h. The bacteria were collected by centrifugation (6000× *g*, 15 min), then lysed via the supersonic schizoanalysis method and analyzed by SDS-PAGE. Protein was purified by the Ni-NTA His•Bind^®^ Resin system (Transgentech, DP101, Beijing, China) and used to immunize two rabbits to produce anti-MCP polyclonal antibody. Indirect immunofluorescence assay (IFA) of 293T cells transfected with pcDNA3.1-MCP plasmid and Western blot (WB) was used to test the reactivity of the antibody.

### 2.8. Indirect Immunofluorescence Assay (IFA)

293T cells transfected with pcDNA3.1-MCP recombinant plasmid were washed twice with phosphate-buffered saline (PBS) and fixed with acetone. Rabbit anti-MCP polyclonal antibody (1:1000 dilution), or preimmune rabbit serum (1:1000 dilution), was added over the cells and incubated for 1 h at 37 °C. Cells were then washed thrice with PBS, and Alexa Fluor 488-conjugated anti-rabbit IgG (Thermo Fisher Scientific, Waltham, MA, USA) at a 1:1000 dilution was then added. After 30 min of incubation at 37 °C, the cells were washed thrice with PBS followed by 4′,6-diamidino-2-phenylindole (DAPI) staining and were visualized under a fluorescence microscope.

### 2.9. Western Blot

The purified MCP protein, 293T cells transfected with pcDNA3.1-MCP recombinant plasmid and concentrated virus inoculum samples were separated by 12% SDS-PAGE. The proteins were transferred onto a polyvinylidene difluoride (PVDF) membrane that was subsequently blocked with Tris-buffered saline (TBS) containing 3% bovine serum albumin (BSA) at room temperature for 2 h. Proteins were detected using the anti-MCP antibody at 1:1000 dilution followed by incubation with horseradish peroxidase (HRP)-conjugated anti-rabbit IgG (1:5000 dilution; Thermo Fisher Scientific).

### 2.10. Animal Experiment

The virus inoculum used for infection trials was prepared as described in [16] with some modifications. Briefly, the spleen and kidney tissue were removed from naturally infected large yellow croakers. These tissues were homogenized in Hanks balanced salt solution (HBSS) and centrifuged at 4000× *g* for 30 min at 4 °C. The supernatant was filtered through a 0.45 μm membrane filter and further concentrated by ultracentrifugation through a 20% (wt/vol) sucrose cushion (140,000× *g* for 6 h). Finally, the precipitate was resuspended with HBSS, and virus inoculum was quantified by qPCR. There were 2 × 10^9^ copies of viral genomes in 100 μL of virus stock.

Healthy large yellow croakers (15 ± 3.5 g) from the Zhoushan Fishery Research Institute, Zhejiang, China, were maintained at 27 °C in a flow-through water system for one week before experimentation. Two hundred large yellow croakers were randomly separated into four groups. Two groups of fish, with 50 fish in each group, received 100 μL virus inoculum by intraperitoneal injection for survival evaluation and samples collection, respectively. Two groups of fish, challenged with equal volume of HBSS, were used as the control. Fish were monitored three times daily for mortality. Dead fish were removed and average cumulative mortality was calculated.

Due to the fact that large yellow croakers are prone to death when they are reared in laboratory conditions, the trial lasted 14 days. Five fish from the challenged and HBSS groups were randomly selected and euthanized at 3, 5, 7, 10, and 14 days post-injection. For viral load determination in specific tissues, liver, intestine, spleen, kidney, gill, and head kidney were harvested. Tools used for sample collection and processing were carefully cleaned with 75% ethanol between each sample to prevent cross-contamination. For viral load determination in specific tissues, all the samples were weighed and homogenized in 1 × PBS by bead beating using sterile zirconium oxide beads (MidSci, St. Louis, MO, USA). Total DNA was extracted from tissue homogenates and tested by qPCR analysis targeting the LYCIV ORF 21 gene, as described above. Virus genome copy number is calculated as per milligram (mg) of tissue samples.

## 3. Results

### 3.1. Identification of the Etiological Agent

Selected fish from the cage-cultured large yellow croaker farms in Zhoushan island, China that exhibits obvious symptoms were firstly examined for parasitological infections of the gills, spleen, kidney, and intestine. No relationship between the clinical signs and parasitic infestation could be shown. Subsequently, bacteriological examinations of *Pseudomonas plecoglossicida*, *Aeromonas hydrophila*, *Vibrio alginolyticus*, *Vibrio harveyi* were performed using methods described previously [17,18], and none of these pathogenic bacteria were isolated from the infected fish. 

Since we could not identify the pathogen, the spleen tissues were subjected to the EM. According to EM, numerous viral particles were in the cytoplasm. The virus particles were 150–200 nm in diameter with an electron-dense core (Figure 1). Subsequently, the gills, spleen, and kidney collected from the same fish were mixed together for further virus detection by qPCR or RT-PCR assays with specific primers for large yellow croaker iridovirus, nervous necrosis virus, picornavirus, and infectious pancreatic necrosis virus as described previously [19,20,21]. The results showed that three out of six infected fish were positive for LYCIV with Ct values of 28.11, 23.76 and 31.30, respectively. The Ct values of the other three samples were 0, 36.76 and 37.84.

### 3.2. Full-Length Genomic Sequencing and Phylogenetic Analyses

The full-length genome sequences were determined by next-generation sequencing and deposited into Genbank under accession no. MW139932. The genome is presented in its full length of 112,043 bp with a G + C content of 53.5% (Table 1). A total of 120 ORFs were identified coding proteins ranging from 45 to 1169 amino acids on the sense (R) and antisense (L) DNA strands. Among the 120 putative genes, 43 were highly homologous to those of other iridoviruses and have defined functions (Figure 2). 

Results of multiple alignments showed that the genome sequence of the virus being studied most closely resembles Pompano iridoviruses isolated in the Dominican Republic with the pairwise nucleotide identities being 99.3% (Table 1). The genome sequence similarity between our virus and the large yellow croaker iridovirus isolated in China in 2003 was only 98.4% (Table 1). Sequence differences between our virus and species belonging to other genera (i.e., *Ranavirus*, *Lymphocystis*, *Chloriridovirus*, *Decapodiridovirus*, *Iridovirus*) range from 39.5 to 63.4%, making it easy to distinguish between them (Table 1). 

To further examine the relationships with other *Iridovidae* family members, phylogenetic analyses were performed on the full-length genome sequences of iridoviruses available from the Genbank database. Our virus was phylogenetically closely clustered with the Pompano iridovirus isolated in the Dominican Republic, forming a sublineage of the genus *Megalocytivirus* including other marine fish iridoviruses isolated in Asia (Figure 3). We named this novel virus the large yellow croaker iridovirus ZS-2020 (LYCIV-ZS-2020).

### 3.3. Prokaryotic Expression and Purification of MCP and Generation of Specific Antibody

The expression of the full-length recombinant MCP protein fused with a 6 × His tag in *E. coli* was subjected to sodium dodecyl sulfate polyacrylamide gel electrophoresis (SDS-PAGE) and Western blot (WB) analysis. The apparent molecular mass of the MCP protein was approximately 50 kDa (Figure 4A). Subsequently, the purified protein was used to immunize New Zealand White Rabbit to produce anti-MCP polyclonal antibodies. The antibody could recognize MCP protein in the WB assay, as it reacted strongly with recombinant full-length MCP protein, pcDNA-3.1-MCP recombinant plasmid-transfected cell lysate, or virus-infected large yellow croaker spleen tissues lysate (Figure 4B). Immunofluorescence assay (IFA) revealed specific fluorescence in pcDNA-3.1-MCP recombinant plasmid-transfected cells, whereas no signal was detected in the transfected cells with the preimmune serum (Figure 4C).

### 3.4. Pathogenicity of the LYCIV-ZS-2020 in Large Yellow Croaker

During the animal experiment, clinical signs such as dark color, enlarged spleen and kidney, and pale gills occurred from 4 days post-injection and reached a peak at 7 days post-injection in fish infected with the LYCIV-ZS-2020. 

Groups infected with the LYCIV-ZS-2020 showed cumulative mortality of 52%, with death primarily from 4 to 9 days post-injection. In the mock-infected group, one fish died on 9, 10, and 12 days post-injection (Figure 5A). We examined the presence of the virus genome in the tissue samples collected from the dead fish of the LYCIV-infected group and mock-infected group. Relative high doses of virus genome were observed in the spleen and head kidney tissues in dead fish of the LYCIV-infected group. All the samples were negative for LYCIV in dead fish from the mock-infected group (Appendix A). Analysis of tissue samples by qPCR showed that LYCIV was detected in the spleen, kidney, gill, and head kidney of LYCIV-infected fish but not the mock-infected group (Figure 5B–E). LYCIV was mainly detected in the spleen and head kidney, with a continuous and decent amount of detectable viral genome copies up to 14 days post-injection (Figure 5B,C). In the gills, while viral genomes were hardly detected in the early stages of infection (from 3 to 5 days post-injection), more viruses were detected from 7 to 14 days post-injection. Interestingly, when compared to the gills, the virus genome level had an opposite trend in the kidney. Viral genome was detected modestly in the kidney early after intraperitoneal injection, which then declined and reached undetectable levels from 7 or 14 days post-injection. Only very low levels of viral genome were detected in the intestine from 3 to 10 days post-injection, and no virus was detectable at later time points in the intestine. In the liver, the viral genome was either undetectable or at extremely low levels. In the mock-infected group, the virus could hardly be detected (Appendix A).

## 4. Discussion

In this study, we obtained the full-length genome sequence of LYCIV-ZS-2020, which is the second full-length genome sequence of large yellow croaker iridovirus reported so far. The sequence similarity between LYCIV-ZS-2020 and the first reported large yellow croaker iridovirus is only 98.4%, suggesting that there are many differences between the two. 

Our artificial infection trial showed that the cumulative mortality in the LYCIV-infected group was 52%, indicating LYCIV-ZS-2020 is pathogenic to large yellow croakers to some extent. The mortality observed in natural infection is much higher than in our artificial infection experiment, which may be a result of potential co-infection with some unknown pathogen in seawater.

By using qPCR, we quantitatively monitored the dynamic distribution of iridovirus in various tissues of large yellow croakers. Our results suggested that the head kidney, gill and spleen of large yellow croaker are likely the main target organs for LYCIV-ZS-2020. This is consistent with clinical symptoms such as abdominal bulging, ascites, white gill filaments, and obvious enlargement of the spleen. However, it is worth noting that the virus in the gills was not detected until 7 days post-injection, which may be a result of inoculation through intraperitoneal injection. The gill is a vital organ in a fish’s blood circulation system where the exchange of oxygen, nutrient, and waste takes place. We speculate that the constant and ample blood flow through the gill may contribute to the spreading of the virus to the gill from immune organs, such as the spleen, kidney, and head kidney, where the virus actively replicates. Once the virus reaches gill tissue and replicates, it can cause tissue lesions and eventually leads to atrophy and whitening of the gill filaments. Another possibility that may lead to gill whitening is an infection of the kidney and spleen severely reducing the number of red blood cells. In the head kidney, a relatively high dose of the virus could be detected even at 14 days post-injection, the most likely explanation is that redistribution of the original inoculum from other organs into the head kidney, with eventual degradation over time. 

One interesting finding that cannot be overlooked is that trace amounts of the virus were detected in the intestine sample of LYCIV-infected fish up to 10 days post-injection. If LYCIV were indeed inside the digestive tract, then we cannot rule out the possibility that the virus excreted with feces might have contaminated the water, which in turn caused lesions in the gill tissue of fish that are already infected. However, detection of LYCIV in the intestine sample may also be a result of the virus adhering to the digestive tract during the injection. Therefore, further studies are needed to investigate if LYCIV-ZS-2020 is able to escape host immunity and enter the digestive tract.

Compared with the pathogenicity of other members of the *Megalocytivirus* genus [22,23], it is possible that LYCIV-ZS-2020 was spread to naturally infected large yellow croakers from other fish species, and it has not yet fully adapted to the new host, causing it to replicate poorly in the cells of large yellow croaker. Members of the genus *Megalocytivirus* are known to have broad host tropism. Large yellow croaker farmers often use other fish caught in the ocean as a source of feed, which increases the possibility of interspecies transfer of iridovirus. However, there is only a limited number of iridovirus genome sequences that are publicly available, making it difficult to study the origin of LYCIV-ZS-2020 based on molecular genetic evolution.

## 5. Conclusions

In summary, we discovered a novel large yellow croaker iridovirus and conducted an artificial infection trial. This is the first study to directly prove the pathogenicity of large yellow croaker iridovirus, which provides valuable experimental evidence to solve the controversy over the pathogenicity of this virus.

## Figures and Tables

**Figure 1 viruses-14-00208-f001:**
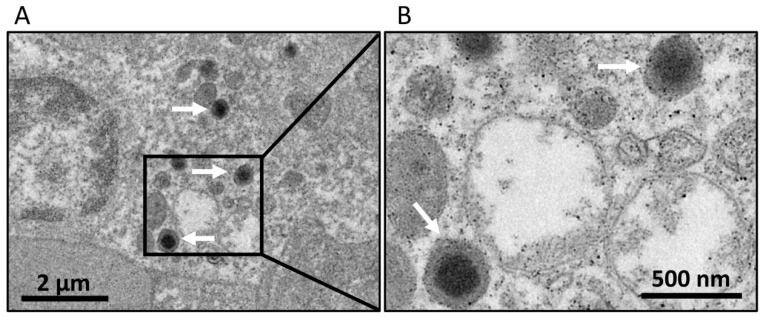
Electron micrograph of viral particles in the cells of spleen tissues. (**A**) Numerous viral particles are in the same cell. (**B**) The outer membranes and central electron-lucent core of mature virions are visible in the enlarged image. Black square area of (**A**) is enlarged in the (**B**). White arrow indicating the virus particle.

**Figure 2 viruses-14-00208-f002:**
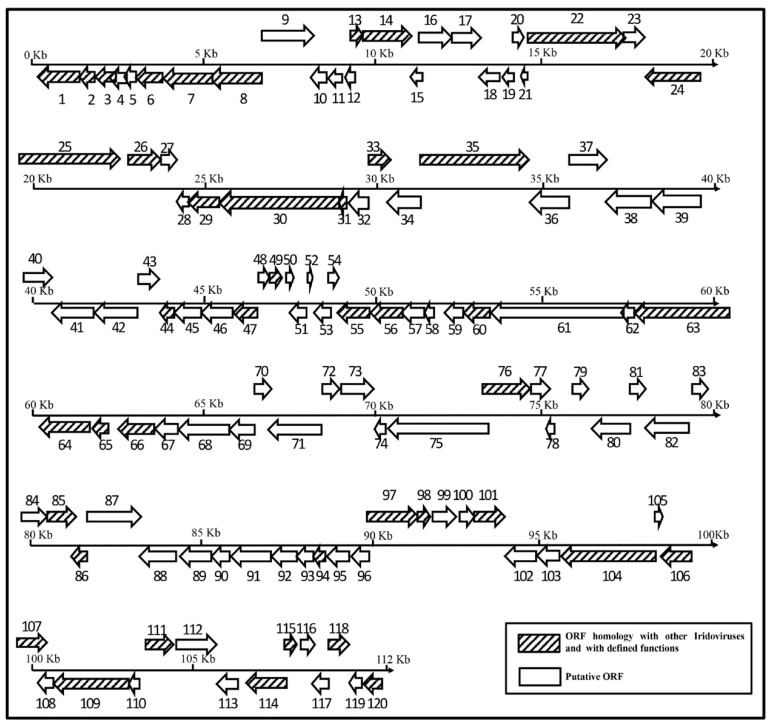
Organization of the LYCIV-ZS-2020 genome. Predicted ORFs are numbered from left to right and represented by arrows indicating their approximate size, location, and orientation based on the positions of methionine start and stop codons. Arrows with diagonals represent the ORFs homology with other iridoviruses and defined functions, and the white arrows represent those with unknown functions.

**Figure 3 viruses-14-00208-f003:**
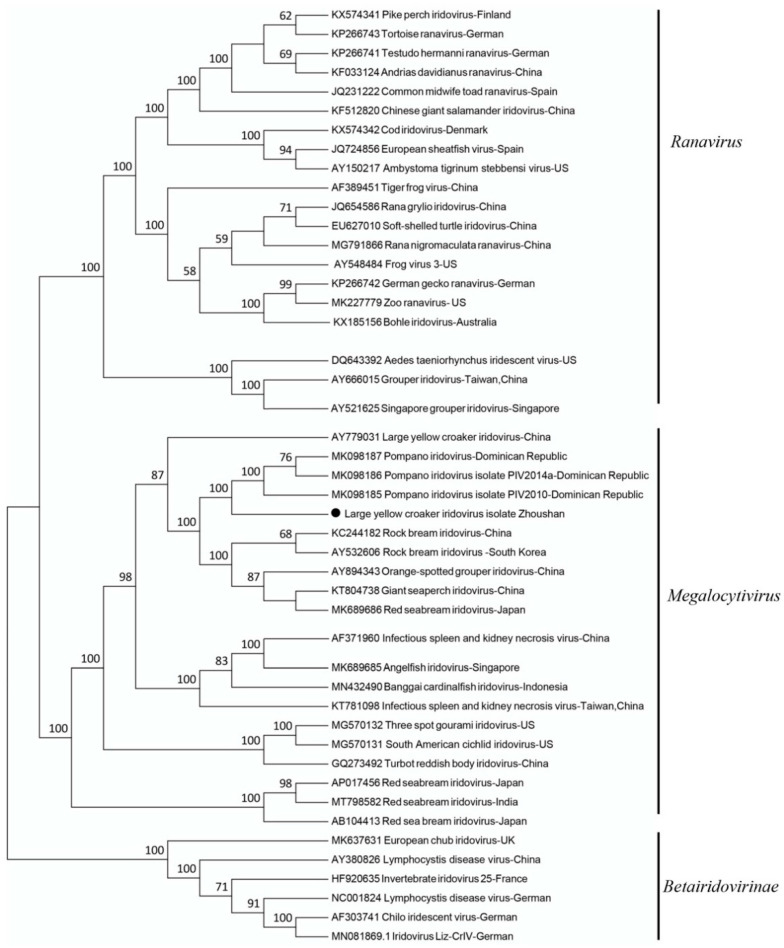
Phylogenetic analysis of LYCIV-ZS-2020 and the other representative iridoviruses based upon nucleotide sequences of the full-length genome. The tree was constructed by the neighbor-joining method. Bootstrap values are indicated for each node from 1000 resamplings. The names of the viruses, GenBank accession numbers, and locations are shown. The solid black circle indicates the LYCIV-ZS-2020 reported in this study.

**Figure 4 viruses-14-00208-f004:**
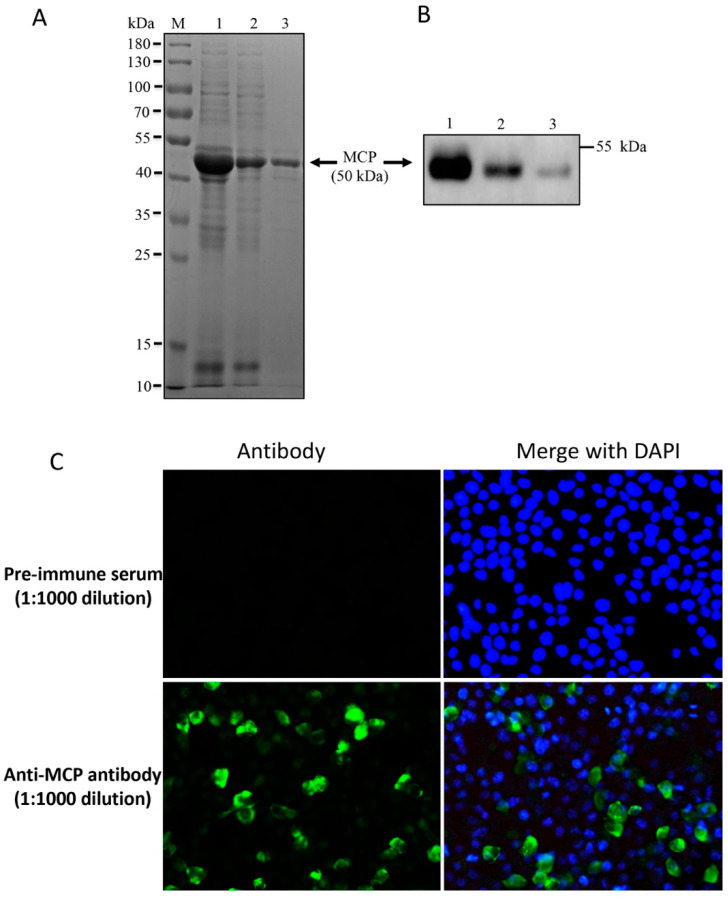
Expression of the LYCIV MCP protein and antibody preparation. (**A**) SDS-PAGE of recombinant MCP expressed in *E. coli*. Rosetta (DE3) cells containing pET28a-MCP were induced with IPTG and then subjected to supersonic schizoanalysis. *E. coli* supernatant (lane 1) and inclusion bodies (lane 2) were harvested, followed by the purification of precipitates (lane 3). M: Molecular marker. (**B**) Western blot analysis using an anti-MCP antibody (1:1000); lane 1: recombinant full-length MCP protein; lane 2: lysate from pcDNA-3.1-MCP recombinant plasmid-transfected cell; lane 3: virus-infected large yellow croaker spleen tissues lysate. (**C**) MCP proteins were detected by IFA (green) with anti-MCP antibody (1:1000) in pcDNA-3.1-MCP recombinant plasmid-transfected cells at 24 h post-transfection. The nucleus were staining with DAPI (blue).

**Figure 5 viruses-14-00208-f005:**
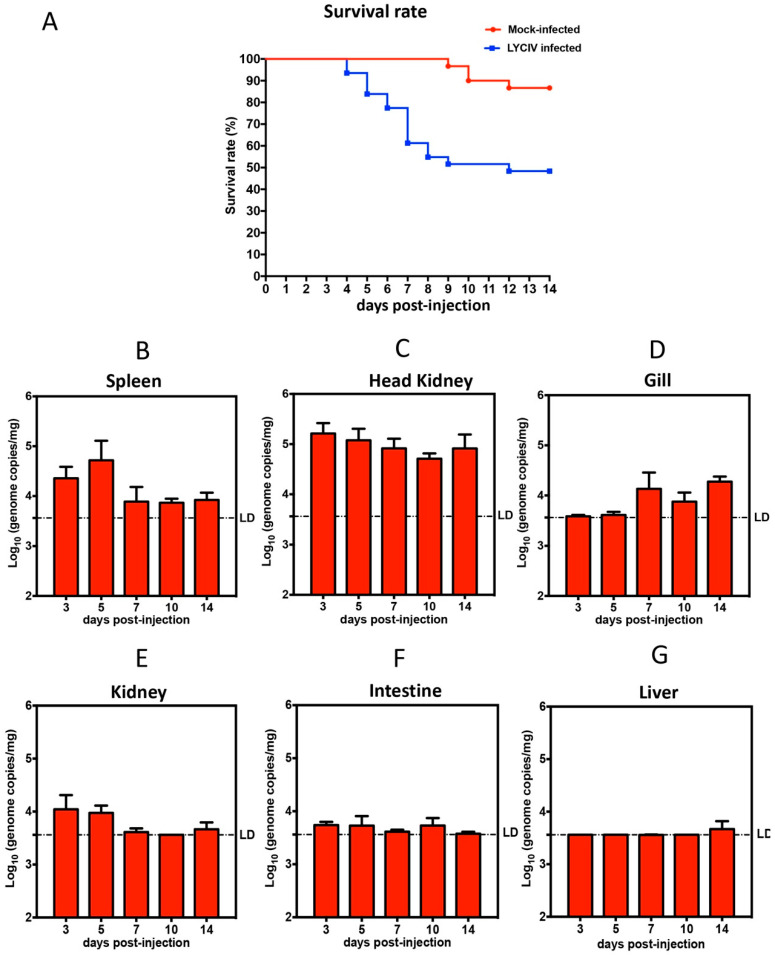
Pathogenicity of LYCIV-ZS-2020 on large yellow croaker. Large yellow croakers were infected intraperitoneally with LYCIVor HBSS. (**A**) The survival rate was calculated daily until 14 days post-injection for large yellow croakers infected with LYCIV (blue line) or HBSS (red line). Viral loads in different tissue samples, including spleen (**B**), Head kidney (**C**), gill (**D**), kidney (**E**), intestine (**F**), and liver (**G**) collected at 3, 5, 7, and 14 days post-injection were determined by qPCR and expressed as virus genome copies per milligram (mg) tissue sample. The limit of detection was 3.4 × 10^3^ genome copies/mg. The error bars represent the standard deviation of the data. The blacked dotted line indicating the limit of detection (LD).

**Table 1 viruses-14-00208-t001:** Nucleotide identities for the full-length genome of LYCIV-ZS-2020.

Genus	Strain	Similarity with LYCIV-ZS-2020	GenBank Accession No.	G + C
*Megalocytivirus*	Large yellow croaker iridovirus-ZS-2020	100	MW139932	53.5
*Megalocytivirus*	Pompano iridovirus	99.3	MK098187	53.4
*Megalocytivirus*	Giant seaperch iridovirus	98.8	KT804738	53.0
*Megalocytivirus*	Orange-spotted grouper iridovirus	98.8	AY894343	53.0
*Megalocytivirus*	Rock bream iridovirus	98.7	KC244182	53.0
*Megalocytivirus*	Large yellow croaker iridovirus	98.4	AY779031	53.9
*Megalocytivirus*	Turbot reddish body iridovirus	96.6	GQ273492	55.0
*Megalocytivirus*	South American cichlid iridovirus	95.9	MG570131	56.0
*Megalocytivirus*	Infectious spleen and kidney necrosis virus	92.8	AF371960	54.8
*Megalocytivirus*	Red seabream iridovirus	74.8	MT798582	53.0
*Ranavirus*	Cod iridovirus	63.4	KX574342	54.9
Ranavirus	Tortoise ranavirus	63.1	KP266743	55.2
*Ranavirus*	Chinese giant salamander iridovirus	62.7	KF512820	55.2
*Ranavirus*	Frog virus 3	62.6	AY548484	55.0
*Ranavirus*	European sheatfish virus	61.6	JQ724856	54.0
*Ranavirus*	Singapore grouper iridovirus	59.7	AY521625	48.0
*Lymphocystivirus*	Lymphocystis disease virus type 1	57.3	NC001824	29.1
*Lymphocystivirus*	Lymphocystis disease virus type C	44.6	AY380826	27.0
*Chloriridovirus*	IIV 3	52.3	DQ643392	47.9
*Chloriridovirus*	IIV 22	42.1	HF920633	28.0
*Chloriridovirus*	IIV 25	40.8	HF920635	30.3
*Decapodiridovirus*	Shrimp hemocyte iridescent virus	48.4	MF599468	34.6
*Iridovirus*	IIV 31	38.3	HF920637	35.1
*Iridovirus*	IIV6	39.5	AF303741	28.6

## Data Availability

Not applicable.

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
