# Peer review of "Genetic and Pathogenic Characterization of a New Iridovirus Isolated from Cage-Cultured Large Yellow Croaker (Larimichthys crocea) in China"

_viruses, 2022, doi:10.3390/v14020208_

Round 1
Reviewer 1 Report
Reviewer comments: Manuscript ID: viruses-1537973
Type of manuscript: Article
Title: Genetic and pathogenic characterization of a new iridovirus isolated from cage-cultured large yellow croaker (Larimichthys crocea) in China Special Issue: Emerging Viruses in Aquaculture.
The work is an important contribution to improve the current knowledge of iridovirus and its pathogenicity in Large yellow croaker (Larimichthys crocea), so economically important farmed species in China.
This work completes your group's previous investigation “Complete Genome Sequence and Phylogenetic Analysis of Red Seabream Iridovirus Isolated From a Cage-Cultured Small Yellow Croaker (Larimichthys Polyactis) in China” by Wang G. et al, Research Square, 23 Nov 2021, and confirms that the genus Larimichthys is sensitive to iridoviruses.
A greater knowledge of the pathogenicity of this virus opens the possibility in the future to test vaccines for the genus Larimichthys, as already in use for other species.
Certainly this virus will have to be considered in the repopulation programs of this species in the wild, as shown by recent works “Could the wild population of Large Yellow Croaker Larimichthys crocea (Richardson) in China be restored? A case study in Guanjingyang, Fujian, China”, by Guanqiong Ye et al., Aquat. Living Resour. 2020, 33, 24.
The manuscript looks interesting, the investigation is well set up, rigorous and complete.
The bibliography is extensive and updated.
The figures/tables/ phylogenetics trees are well made and easy to understand.
Some considerations:
Animal experiment: usually it is used to make the experimental infections in triplicate, in this case instead the duplicates have been made. In any case, the large number of fish used is comforting.
Histological images: very interesting. It would have been interesting to show photos at higher magnification to appreciate lymphocytic infiltrations or necrosis that usually appear in iridovirus infections, or were they absent?
Anyway congratulations, it's a great job.
I have no major revisions to propose, only minor revisions.
Minor revisions:
line 516… diffcult to study….please change in…. difficult to study;
Line 577: Psetta maximus…please change in …. Psetta maxima (compare with Fishbase.de)
Reviewer 2 Report
This manuscript describes the use of several methods to increase our understanding of large yellow croaker iridovirus (LYCIV). The writing is generally good. However, the information provided contains significant discrepancies and gaps that, in total, do not strongly support the authors conclusion that LYCIV is pathogenic to large yellow croaker. I offer several specific comments that might help clarify the presentation and fill some of these gaps:
Line 67 – this should be “large yellow croaker iridovirus (LYCIV)” [‘croaker’, not ‘croakers’]
Lines 86 – 88, “In this study, we identified the causative pathogen as a new iridovirus through transmission electron microscopy (TEM) and next-generation sequencing.” TEM and sequencing are good methods for identifying a virus, but they are not sufficient to demonstrate the cause of a disease; therefore, this sentence needs to be rewritten; perhaps something like, “In this study, we identified a new iridovirus in these fish through transmission electron microscopy (TEM) and next-generation sequencing.”
Line 143 – Clarification is needed on how the copy number is expressed. Is it copy numbers per g of sample? per g of DNA?
Line 253 – injection of viral material does not necessarily mean that infection will occur. Therefore, I recommend replacing “post-infection” with something like “postexposure” or post-injection.” Also, more information is needed here on how viral load was determined (qPCR?).
Scientific writing concept - Describe findings, not what was done to obtain the results. For example, rewrite sentences to avoid words like “was observed” (Line 62) or “were observed” (Line 64) that do not contribute to the description. For more direct wording, replace 'were observed' with 'occurred', or simply delete ‘observed.’ Consider also revising the use of “observed” (e.g., line 277). The ‘find’ feature of a word processing program can be used to find and correct all instances throughout the paper.
Lines 253 – 255 – Were dissection tools cleaned between harvesting of each organ for each fish (this is not common)? If not, then the results cannot be used to confidently support what is described in line 385 (i.e., “To determine the target tissues of the virus…”), particularly when many of the determinations seem to be very close to the limits of detection. Instruments used to harvest one organ might have carried detectable particles to the next harvested organ. Likewise, this information is important for putting the conclusion expressed in lines 474 - 475 in perspective (“By using qPCR, we quantitatively monitored the dynamic distribution of iridovirus in various tissues of large yellow croaker.”). Manuscript wording might need to be revised accordingly.
Lines 259 – 263: What negative control was used for the IHC determinations? Were the slides examined while the scientist was blinded to the exposure history of the sample in the slide? IHC commonly produces nonspecific staining; therefore, it is important to include methods to rule out nonspecific staining.
Lines 266 – 284 – these are all methods that can be used to identify potential causes of increased mortality. For future study, I recommend adding histopathology as a step before electron microscopy. Of, course that requires having a skilled pathologist, which is not always available…
Lines 282 – 284, “LYCIV was detected in the 3 out 6 infected fish. Therefore, we named this novel virus as large yellow croakers iridovirus ZS-2020 (LYCIV-ZS-2020).” The logic here is incomplete. Not clear is the jump from LYCIV detected in three fish to calling it a novel virus with a different name. Is this truly being reported as a novel virus, or is it just a genotype of LYCIV? On line 305, the original LYCIV is described as a “strain.”
My interpretation of this manuscript is that additional work described in the following sections was important in supporting the conclusion of at least a novel genotype, but that is not obvious to a new reader at this point in the paper. Also, this wording states that six fish were infected, but no evidence is provided that the three fish PCR-negative for LYCIV were infected with anything. Consider rewriting this section to say something like, “PCR tests were positive for LYCIV in the 3 out 6 fish; additional work was done to determine the relation of this virus to previous findings.” After all the evidence is presented, then provide the statement suggesting the name of this virus as large yellow croaker iridovirus ZS-2020 (LYCIV-ZS-2020).
Lines 374 – 377 – The information in these sentences seems to fit better in the Methods section than in the Results section. I suggest moving the information here to the Methods section.
Lines 383 – 385 – The reference to Figure 5 better fits the information provided in line 383; therefore, I recommend moving “Figure 5A” to after ‘dpi’ in line 384.
Sentence spanning lines 385 – 388 – information in this sentence should be moved to the Methods section.
Sentence in lines 388 – 389 – this should be rewritten to report results. Use the Discussion section to describe what the results suggest.
Figures 5B – 5G – I find this presentation confusing. It appears that all of the mock-infected fish had a positive PCR test at ~3.5 Log10 genome copies per gram. I recommend removing the mock infected bars in favor of an explanation in the text. Also, the error bars need to be explained. Finally, this figure provides little evidence to support the hypothesis that the virus replicated in the fish. In only the gill do the copy numbers seem to increase during the experiment, and even there the small copy number might be explained by redistribution from other organs where the copy number seems to decrease over time.
Lines 390 - 391, “Active replication” – A positive PCR result can be a due to things other than active replication (e.g., macrophages ingesting infected cells where viral segments are present but not replicating). Therefore, this wording needs to be revised accordingly (e.g, the PCR tests detected viral segments in certain organs more than others through 14 dpi).
Line 395 – I do not find a description of statistical analysis being done. Therefore, I recommend replacing “significantly” with “nominally.” Also, I recommend including a dataset with all of the PCR results (e.g., in an Excel spreadsheet).
Lines 412 – 414 – Comparison of the gill PCR and IHC results from 5 dpi do not support “Immunohistochemistry assay confirmed the above mentioned results of the virus genome detection.” Indeed, Figure 5D shows no virus in the gill at 5 dpi, whereas Figure 6 seems to show abundant viral antigens in the gill at 5 dpi. See my comment about the IHC methods (Lines 259 – 263); the gill staining in Figure 6 might be nonspecific.
Scientific writing concept - For the Discussion section, I recommend the guide for authors used by the Canadian Journal of Fisheries and Aquatic Sciences. “Limit the Discussion to giving the main contributions of the study and interpreting particular findings, comparing them with those of other workers. Emphasis should be maintained on synthesis and interpretation and exposition of broadly applicable generalizations and principles. If these are exceptions or unsettled points, note them and show how the findings agree or contrast with previously published work. Limit speculation to what can be supported with reasonable evidence.”
Applied to this manuscript, the first three paragraphs are general information. I recommend removing this information from the Discussion and moving the most relevant concepts to the Introduction section or to support discussion of the main contributions of the study and interpreting particular findings. Also, lines 462 – 464 would better fit in the Methods section, and lines 465 – 467 are a repeat of the Results.
Line 472 – good point. Mortality in the field might also be related to interaction with adverse environmental conditions (water quality, crowding, etc.). Also, given that only 3 of 6 outbreak fish were PCR+ for LYCIV, a reasonable conclusion is that something else was causing the mortality among the larval fish.
Lines 488 – 490 – White gills are more likely a result of anemia than replication of the virus in gill tissues. This would also be expected if infection of the hematopoietic tissues of the kidney decreased red blood cell production, or if infected red blood cells were removed in greater numbers from the spleen. This is a good example where histopathology would help with the interpretation of the pathogenesis.
Lines 496 – 497 – the manuscript does not provide strong evidence that the virus “entered digestive tract”; an alternate hypothesis is that small amount of the virus injected into the body cavity simply adhered to the digestive tract.
Lines 501 – 502, “…(the highest viral load was found in kidney tissue, which is only 105.2 copies/g).” – It is not clear how this number is derived from what is shown in Figure 5, where kidney samples have >5 Log10 genomic copies (= >100,000 copies per g). I would not expect any lesions with only 100 copies/g; therefore, this finding supports the conclusion that the virus did not replicate or cause the lesions in these fish. Consider adding a calculation regarding what would have happened to the original 2 × 10^9 copies of viral genome (line 238) once distributed into the organs of the fish; if my calculations are correct, the injected 15-g fish would have started the experiment with about 1.3 × 10^8 viral particles per g of fish tissue. This seems like a high number in contrast to how it is characterized in line 503 as “a low amount of virus” (e.g., I am coauthor of a study published recently, where we injected 10^6 copies of a virus that could not be replicated in cell culture; this virus replicated abundantly in the host).
Round 2
Reviewer 2 Report
Thank you very much for clarifying the presentation and adding a dataset. These additions make clear several things that still need to be addressed.
Line 237 – My understanding of these methods is that they are not specific for LYCIV; if correct, the possibility that the inoculum contained another virus needs to be considered.
Line 254 – A statement is needed here to clarify whether dead fish were tested by PCR for LYCIV. I do not see these results anywhere. This is unfortunate. It is very helpful in these experiments to compare viral loads in dead vs. live fish. Pathogenic viruses tend to occur at greater load among the dead population.
Line 269 – this should be “copy number” [not “copies number”]
Lines 276 – 278: My interpretation of this is that (i) the anti-MCP antibodies were applied to sections from the exposed fish, and (ii) rabbit IgG were applied to the sections from control fish. What is missing is a description of results when (i) anti-MCP antibodies are applied to sections from the control fish, and (ii) rabbit IgG is applied to sections from the exposed fish. This information is needed to rule out nonspecific staining. Without these controls, we cannot be confident in the results. If this work was done, that work needs to be clarified. If the work was not done, the work must be done or the IHC results will need to be removed from the manuscript. I understand from response 17 that PCR was not done on samples from the same fish that were used for IHC; given that we agree that “virus replication has individual variability,” this is further reason that we can have little confidence in the IHC results. Finally, the statement that “The tissue is so small and thus not sufficient for both experiments” is not consistent with lines 251 – 253 (“Before the experiment, gills of 251 three fish from each group were selected and tested negative for iridovirus by qPCR (data not shown).”) Fish have 8 gill arches; it would have been possible to sample two gills from each fish: one for PCR and one for IHC.
Line 297 – What were the loads (or Ct values) of LYCIV in these three fish?
Line 301 – readers will be looking at the image, so I recommend using present tense (“particles are in the same cell”) as in the caption for image B.
Line 393 – “occurred” may be deleted here.
Supplemental file – Thank you for including this. Still missing are the original Ct values from each fish, results from control fish, and any results from testing of dead fish. Please add these data.
Lines 395 – 397 “All the deceased fish in the mock-infected group showed no evidence of virus infection.”: If these fish were not tested by PCR for LYCIV, then this statement needs to be modified to something like, “All the deceased fish in the mock-infected group had no clinical signs of virus infection.” If the fish were tested by PCR for LYCIV, then add the results to the supplementary file.
Lines 397 – 400: now that Fig. 5 is revised, it no longer shows control fish results. Reference can be added to the revised Supplemental file (see previous comment)
Figure 6 caption. If my interpretation is correct that the fish subjected to ICH were not tested for LYCIV, then the PCR results here need to be deleted.
Finally, what evidence do we have the LYCIV replicated in the exposed fish? As I said in my original review, the original 2 × 10^9 copies of injected viral genome (new line 241) once distributed into the organs of the 15-g fish dilute to an average of 1.3 × 10^5 viral particles per mg of fish tissue. If we compare this with the data in the supplemental file “Genome copies per mg tissue,” we find in only the head kidney are the average concentrations ever greater than 1.3 × 10^5 viral particles per mg of fish tissue (day 3 = 1.8 × 10^5); this pattern is better explained by redistribution of the original inoculum from other organs into the head kidney, with eventual degradation over time. Further, mortality first occurs on day 4 PI, but by day 5 the head kidney viral load is decreasing and totals less than the original inoculum.
